# Strength Asymmetries Are Muscle-Specific and Metric-Dependent

**DOI:** 10.3390/ijerph19148495

**Published:** 2022-07-12

**Authors:** Gennaro Boccia, Samuel D’Emanuele, Paolo Riccardo Brustio, Luca Beratto, Cantor Tarperi, Roberto Casale, Tommaso Sciarra, Alberto Rainoldi

**Affiliations:** 1Department of Clinical and Biological Sciences, University of Turin, 10126 Turin, Italy; gennaro.boccia@unito.it (G.B.); paoloriccardo.brustio@unito.it (P.R.B.); cantor.tarperi@univr.it (C.T.); 2NeuroMuscularFunction Research Group, School of Exercise and Sport Science, SUISM, University of Turin, 10126 Turin, Italy; luca.beratto@unito.it; 3Department of Neuroscience, Biomedicine and Movement, University of Verona, 37129 Verona, Italy; samuel.demanuele@univr.it; 4Opusmedica Persons Care & Research, NPO, 29121 Piacenza, Italy; robertocasale@opusmedica.org; 5Joint Veterans Defence Center, Scientific Department, Army Medical Center, 00184 Rome, Italy; sciarratommaso@hotmail.com; 6Department of Medical Sciences, University of Turin, 10126 Turin, Italy

**Keywords:** explosive contraction, muscle quickness, dominance

## Abstract

We investigated if dominance affected upper limbs muscle function, and we calculated the level of agreement in asymmetry direction across various muscle-function metrics of two heterologous muscle groups. We recorded elbow flexors and extensors isometric strength of the dominant and non-dominant limb of 55 healthy adults. Participants performed a series of explosive contractions of maximal and submaximal amplitudes to record three metrics of muscle performance: maximal voluntary force (MVF), rate of force development (RFDpeak), and RFD-Scaling Factor (RFD-SF). At the population level, the MVF was the only muscle function that showed a difference between the dominant and non-dominant sides, being on average slightly (3–6%) higher on the non-dominant side. At the individual level, the direction agreement among heterologous muscles was poor for all metrics (Kappa values ≤ 0.15). When considering the homologous muscles, the direction agreement was moderate between MVF and RFDpeak (Kappa = 0.37) and low between MVF and RFD-SF (Kappa = 0.01). The asymmetries are muscle-specific and rarely favour the same side across different muscle-performance metrics. At the individual level, no one side is more performative than the other: each limb is favoured depending on muscle group and performance metric. The present findings can be used by practitioners that want to decrease the asymmetry levels as they should prescribe specific exercise training for each muscle.

## 1. Introduction

According to the dynamic dominance models [1], the dominant limb might be specialized for controlling movements through predictive mechanisms that are most effective under stable mechanical conditions, while the non-dominant limb might be specialized for impedance control, which imparts stability when mechanical conditions are unpredictable [2]. Although this model does not mention muscle strength or power as critical factors in dominance differentiation, many studies investigated the effect of limb dominance on such muscle performance.

Handgrip muscles are the only ones that present a clear trend (at least in right-handed individuals) of 8–16% towards a stronger dominant than non-dominant side [3]. The effects of limb dominance on other strength tests are not as noticeable. In a recent systematic review, including 19 studies (1880 healthy non-athletes subjects), Kotte et al. [4] reported no difference between the dominant and non-dominant side in elbow flexors and extensors. Ditroilo et al. [5] did not find any difference in maximal voluntary force (MVF) and rate of force development (RFD) between dominant and non-dominant knee extensors in a sample of 152 people of various ages. In sports, two meta-analyses found that lower limb dominance did not influence isometric and dynamic muscle strength [6,7].

The fact that the dominant limb is not stronger or weaker than the non-dominant one at the population level does not imply that one side would be stronger than the contralateral one at the individual level. Regardless of dominance, the interlimb asymmetry varies depending on the test selected, and, thus, the level of asymmetry may strongly depend on tasks [8]. Numerous recent studies found that the direction of interlimb asymmetry is rarely consistent across tests [8,9]. For example, by comparing the lower limb asymmetry during three jump tests, Bishop et al. [10] found that the levels of agreement across jump tests (considering peak torque and other metrics) were poor. While a metric may favour the dominant limb at an individual level, others may favour the non-dominant one. Those studies reinforce that limb dominance does not represent a strong predictor of strength and quickness produced by a muscle group. However, previous studies investigating the consistency of asymmetry direction focused only on one muscle group/kinetic chain [10]. So far, it is unclear if asymmetry direction is consistent among heterologous muscles. Considering the impact of interlimb asymmetry on sports performance [11] it would be essential to understand if, in the presence of an asymmetry, all muscles of one limb are more performative than the contralateral ones.

Measuring the MVF is the most straightforward assessment to detect interlimb strength asymmetry [12]. Since RFD represents a valid alternative to the classical evaluation of MVF [13], RFD is emerging as a meaningful indicator of interlimb asymmetry [14,15]. RFD represents the derivative of force with respect to time [16] and quantifies a muscle contraction’s explosiveness [13]. RFD is an important neuromuscular variable in time-constrained activities [13,16]. Its relevance has been repeatedly demonstrated in sports [17], ageing [18], and disease contexts [19]. The RFD and MVF rely on partially different physiological determinants; for example, MVF is more related to muscle volume [20,21], while RFD is more related to the rate of motor unit recruitments [22]. For these reasons, MVF and RFD are weakly correlated [20,21], especially when RFD is measured in the early phase of a muscle contraction [13]. Nevertheless, RFD and MVF have in common the fact that they are assessed performing maximal-effort contractions. However, not all daily activities and sports gestures are based on maximal-effort contractions. Most actions are likely based on quick contractions of submaximal intensities: for example, walking, running, passing a ball in soccer, or shooting a free shot in basketball. In this context, the adoption of RFD-Scaling Factor (RFD-SF) has emerged as an informative measure to quantify the neuromuscular quickness of submaximal contractions [23,24,25,26]. Interestingly, RFD-SF is weakly correlated to MVF and also to maximal RFD [27]. The interest in this capacity has increased over the last years [28], and RFD-SF has been widely used to identify interlimb asymmetry [15,29,30]. Together these studies suggest that MVF, RFD, and RFD-SF might provide different and complementary outcomes in the assessment of interlimb asymmetry.

In the present study, we focused on upper limbs (elbow flexors and extensors) because they play a critical role in everyday living and sports context in non-disabled, amputee, and wheelchair users. We firstly aimed to examine if dominance affected muscle function. Secondly, we investigated if muscle function asymmetries are muscle-specific or, conversely, if one side is overall more performative than the other independently of the muscle group. Thirdly, we aimed to investigate if, within each muscle group, asymmetry direction is consistent among various muscle performance metrics (MVF, RFD, and RFD-SF).

To answer the first experimental question, we adopted a linear mixed-effects model analysis, while to answer the second and third experimental questions, we tested the agreement between asymmetry direction among heterologous muscle groups and various performance metrics. We hypothesized that: (1) the dominance did not affect muscle functions (MVF, RFD, and RFD-SF); (2) the asymmetry direction agreement among heterologous muscles (i.e., elbow flexors and extensors) was low; and (3) the asymmetry direction agreement among the metrics adopted (MVF, RFD, and RFD-SF) in homologous muscles was low. Finally, we tested the inter-day repeatability of the custom-made isometric dynamometer adopted in the present study as a secondary objective.

## 2. Materials and Methods

### 2.1. Participants

A total of 55 young (31 males and 24 females, mean age = 30 ± 7 years;) physically active healthy individuals (body mass = 70 ± 9 kg, body height = 1.74 ± 0.17 m, body mass index 23.3 ± 1.6) were recruited for the study. Five of them were left-handed. Inclusion criteria were: being adults (≥18 years of age); and being physically active, i.e., participating in moderate-intensity physical activity at least 150 min/week or vigorous-intensity physical activity at least 75 min/week or an equivalent combination of both moderate and vigorous physical activity. Exclusion criteria were: any upper-limb complaints and general illness in the past six months; any clinical evidence of cardiovascular, neuromuscular, or neurological disorders; and participation in any sports that require extensive asymmetric involvement of the upper limbs (such as tennis, badminton, fencing, etc.). All the participants were informed about the testing procedure and provided their written informed consent before participation in the experiments. Participants were instructed to refrain from performing strenuous physical exercise and consuming caffeine 24 h before the experimental session and completed a socio-demographic questionnaire before the experimental sessions. The study was approved by the Ethical Committee (University of Torino—approval no: 510190) and performed in accordance with the Helsinki Declaration.

### 2.2. Experimental Setup

A picture of the experimental setup is reported in Figure 1A. During the testing, participants were comfortably seated on a bench (seat height = 44 cm) with the left or right upper arm vertically and slightly abducted from the trunk (~15° degree). Each participant’s elbow was flexed at 90° from full extension. Furthermore, the elbow leaned over adjustable support to avoid the force exerted with the shoulder and trunk would transmit through the force sensor. The hand and forearm were oriented in a neutral position. The wrist was aligned with custom-built telescopic support (see Figure 1A) and fixed with nonelastic straps to the arm. The custom-built support was rigidly connected to a strain gauge load cell (Model TF 022, cct transducers, Torino, Italy) to record compression/extension forces. Real-time visual feedback of elbow flexor/extensor forces was provided on a computer screen (size screen 48 cm × 27 cm). The force signals were sampled at 100 Hz and converted to digital data with a 16-bit A/D converter (Forza, OT Bioelettronica, Turin, Italy).

### 2.3. Procedure

All participants completed one experimental session during which the muscle-function assessment was performed for elbow flexion and extension of the dominant and non-dominant limb in randomized, counterbalanced order. The experimental protocol was re-administered to a sub-sample of 15 participants to check its test–retest reliability. The experimenters placed particular attention on avoiding torso and shoulder movement during contractions execution. In addition, participants were instructed to avoid trapezius activation during the elbow flexion and body leaning forward during elbow extension. The same investigators conducted all test sessions. A rest of 5 min was observed between testing each muscle group. For each muscle group and limb, the protocol comprised: (1) a warm-up consisting of 10 submaximal isometric contractions (at intensities from 20 to 80% of the perceived maximum force); (2) familiarisation to ballistic contractions (see later); (3) two maximal voluntary isometric contractions; and (4) RFD-SF protocol.

Two 5 s maximal voluntary contractions, interspersed by 2 min of rest, were performed to measure MVF. A third maximal voluntary contraction was performed when the MVF difference between the two trials was higher than 5%. Participants received standardized verbal encouragements during the execution of maximal voluntary contractions.

The RFD-SF protocol started 2 min after the last maximal voluntary contractions. The original RFD-SF protocol requires the performance of 125 ballistic isometric contractions across a full range of submaximal amplitudes [23]. As a reduced form of the original protocol consisting of at least 36 contractions showed reliable results [31], participants were instructed to perform 12 ballistic isometric contractions (interspersed by 5 s) at 80%, 60%, 40%, and 20% of their MVF for a total of 48 contractions (see Figure 1B). They were asked to produce rapid contractions with peak forces reaching approximately ±10% range of the target force. Each pulse was controlled by standardized acoustic cues. When it was obvious that a ballistic isometric contraction had not been performed properly, the same was repeated. The range force was displayed on the computer screen as a horizontal band of 20% MVF width. Participants were explicitly instructed to produce each isometric torque pulse as quickly as possible and then relax instantly. The emphasis was on the quickness of the contraction rather than the accuracy.

### 2.4. Mechanical Signals

Signal processing was conducted using a custom-written software in MATLAB R2020b (The MathWorks Inc., Natick, MA, USA). MVF and RFDpeak were calculated over the raw force signal. MVF was computed as the 0.5 s epoch with the highest value of the force signal. To obtain the RFDpeak, we averaged the three contractions showing the highest maximal RFD (calculated as the peak of the first derivative of force signal among the explosive contractions of the RFD-SF protocol).

To calculate the RFD-SF, the force signal was firstly pre-processed using an overlapping moving window of 0.1 s [5,26,32]. The adoption of a moving window was preferred to a 5 Hz low-pass filter because it does not introduce aberration in the signals (typically evident as a force signal below zero just before the contraction onset). If any countermovement was evident (i.e., a drop in force greater than 0.25 kg in the 250 ms before the contraction onset), the contraction was rejected from the analysis. Then, the first derivative of the force signal was computed to obtain the RFD signal. For each ballistic contraction, peak force and RFDpeak (which is the local maximum of the RFD signal) were calculated.

The RFD-SF was calculated as the slope of the linear regression between peak force and peak RFD obtained in each contraction (Figure 1C). RFD-SF represents how RFD scales with force in a range of submaximal contraction and, thus, quantify the quickness across a span of intensities. The R^2^ was also quantified as it reveals the consistency and linearity of the linear regression. Outliers were detected and removed using the Cook distance methodology to improve the fit of the linear regression [33].

### 2.5. Statistical Analysis

Descriptive data of the dependent variables are presented as mean and standard deviation (SD). The bilateral asymmetry index was calculated for each parameter according to the following formula [34]:(1)$$\frac{\text{Dominant limb}-\text{Nondominant limb}}{\text{Dominant limb}+\text{Nondominant limb}}\times 100$$

We calculated the smallest worthwhile change (SWC, 0.2 × pooled SD [35] to interpret interlimb difference that exceeded this threshold as a true difference [36]. Participants were considered *symmetric* when the interlimb difference was less than SWC [37]. Otherwise, they were considered *asymmetric*, favouring either the dominant or the non-dominant side. Then, Kappa coefficients were calculated to determine the levels of agreement for the direction of asymmetry among muscle groups and performance metrics at the individual level [9]. The Kappa coefficient describes the proportion of agreement between two methods after any agreement may have occurred by chance [38]. We adopted linear weighting kappa [39] to account for how far apart two categories might be (e.g., “asymmetry favouring the dominant limb” is a category closer to “symmetry” than to “asymmetry favouring the non-dominant limb”). Kappa values were interpreted as follows [40]: 0.01–0.20 = slight; 0.21–0.40 = fair; 0.41–0.60 = moderate; 0.61–0.80 = substantial; and 0.81–0.99 = nearly perfect. High Kappa values would mean that the direction of asymmetry tends to be the same for different muscle groups or metrics. Therefore, Kappa statistics were applied to test the dominant vs. non-dominant advantage across muscle groups (considering the same performance metric) and across metrics (considering the same muscle group).

To check if dominance affected muscle function at the sample level (i.e., collectively considering all participants), we performed multilevel mixed-effect linear regression analysis [41]. The adoption of mixed-effects models is essential to account for the fact that each subject was measured four times (i.e., two muscle groups of two sides). Therefore, we considered the dominance and muscle group over participants as random factors. Then we considered dominance, muscle group, and gender as fixed effects.

For each muscle function metric (MVF, RFDpeak, and RFD-SF), intraclass correlation coefficient (ICC), standard error of measurement (SEM), and coefficient of variation (COV) were calculated to assess the interday reliability [42]. According to Koo and Li [43], ICC reliability were interpreted as: >0.90 = excellent, 0.75–0.90 = good, 0.50–0.75 moderate and < 0.50 poor. COV values < 10% were deemed acceptable.

Statistical analysis was performed in R (ver 4.1.1, R Core Team, Vienna, Austria, 2021), the figures were produced using the package ggplot2 [44] and MATLAB R2020b (The MathWorks inc., Natick, MA, USA). The threshold for statistical significance was set at *p* < 0.05.

## 3. Results

The descriptive statistics of the three performance metrics across muscle groups and limb dominance are reported in Table 1. When controlling for gender, the dominant side showed higher MVF (F = 19.2, *p* < 0.001) both in extensors (*p* = 0.010) and flexors (*p* = 0.001) compared to the non-dominant side. The RFDpeak was similar on both sides (F = 3.2, *p* = 0.077), while the RFD-SF was higher in the non-dominant side compared to the dominant side in the elbow extensors (*p* = 0.016) but not in the elbow flexors (*p* = 0.409). Based on SWC analysis (see Table 1), on average more than 65% of individuals were asymmetric.

The distribution of bilateral asymmetry indices is reported in Figure 2A. As can be seen, the distributions are widely distributed both towards the dominant and non-dominant sides. The Figure 2B reports the data of eight representative participants: most individuals show some performance metrics favouring the dominant and some others favouring the non-dominant side. The agreement analysis confirmed this scenario. Indeed, the asymmetry direction agreement between heterologous muscle groups was slight for all metrics: MVF Kappa = 0.13; RFDpeak Kappa = 0.14; and RFD-SF Kappa = 0.16.

The asymmetry direction agreement between muscle performance metrics (among homologous muscle groups) was fair for the agreement between MVF and RFDpeak (Kappa = 0.37), slight for the agreement between RFDpeak and RFD-SF (Kappa = 0.14), and null when comparing MVF with RFD-SF (Kappa = 0.01).

The two most commonly adopted muscle performance metrics, MVF (ICC = 0.93) and RFDpeak (ICC = 0.92), showed excellent reliability, while the RFD-SF showed lower but still acceptable reliability (ICC = 0.69). The coefficient of variation of all variables was <10% (MVF = 7.0%; RFDpeak 6.5%; and RFD-SF 6.9%). The R^2^ of the RFD-SF protocol was on average ≈0.96 for both elbow flexors and extensors.

## 4. Discussion

We measured three muscle-performance metrics (i.e., MVF, RFDpeak, and RFD-SF) in two muscle groups of the upper limbs (i.e., elbow flexors and extensors) of the dominant and non-dominant sides. The main findings were that (1) at the population level, the difference between the dominant and non-dominant side was trivial, when present, and it was favouring the non-dominant side; (2) the asymmetry direction agreement between heterologous muscle groups were relatively poor for all metrics (all Kappa values ≤ 0.16); and (3) the asymmetry direction agreement between muscle performance metrics (among homologous muscles) was moderate between MVF and RFDpeak (Kappa = 0.37) and low between MVF and RFD-SF (Kappa values = 0.01). Overall, the present findings suggest that no one side is more performative than the other: the objectively better side depends on muscle group and performance metrics adopted.

At the sample level (i.e., collectively considering all participants), the MVF was the only metric that showed a clear trend favouring one side compared to the other one. Indeed, in our sample, the MVF was higher on the non-dominant compared to the dominant side (Table 1). This is partially in conflict with previous data showing slightly higher strength on the dominant compared to the non-dominant side [4]. As we did not record any physiological measures of muscle activation or contractile properties, it is impossible to ascribe the side-by-side difference to central or peripheral properties. However, the relatively small sample size of our study (55 subjects) does not allow for inferring the present finding to the general population. Furthermore, the magnitude of difference between the sides was negligible as it ranged from 3 to 6%. A symmetry index lower than 10% is usually considered negligible [6,7]. Nevertheless, the asymmetries directions were sparse at the individual level (see Figure 2). Therefore, the most important findings of the present study regard the analysis of asymmetry direction agreement at the individual level.

For each performance metric, the bilateral asymmetry index at the individual level was muscle-specific. The low Kappa values (all Kappa coefficients were ≤0.16) showed that the between-muscle agreement of asymmetry direction was poor for each metric. Kappa values close to 0 suggest that the direction agreement between the two heterologous muscles was due mainly to chance. Therefore, if one metric favoured the dominant limb in one muscle group (e.g., elbow flexors), this does not necessarily occur for the heterologous muscle (i.e., elbow extensors). As a consequence, it is possible to suggest that participants do not have an overall stronger or quicker side. They have a muscle-specific level of strength and quickness instead. This may be due to differences in morphological (muscle size and architecture) [45] and neural activation features [46] across muscles. Bishop and colleagues [9] highlighted the importance of reporting the agreement between asymmetry direction, as they noted that most previous studies did not mention the direction agreement among asymmetry metrics. Here we expand previous literature demonstrating that, even when assessing the same performance metric, the upper limb’s heterologous muscles do not share the same asymmetry direction.

Asymmetries rarely favoured the same side when considering different performance metrics of homologous (contralateral) muscles. Except for the agreement between MVF and RFDpeak (which was moderate), all other Kappa values were low (≤0.14). RFDpeak has been previously reported to be more sensitive to detect asymmetry than MVF [14]. Here we expand previous literature by demonstrating that RFD and MVF do not point necessarily in the same direction. Furthermore, when comparing MVF with quickness metrics (i.e., RFD-SF), the asymmetry agreement was null (Kappa values = 0.01). RFD-SF is believed to quantify contraction quickness independently by maximal strength [26]. The present findings suggest that at the individual level, maximal strength asymmetry is unrelated to quickness asymmetry. The differences in underlying physiological mechanisms could be responsible for the observed poor agreement [21,47]. From a practical point of view, this finding advocate that practitioners should include muscle-performance tests and metric specifically oriented to muscle quickness, such as relative RFD-SF [48].

The task-specific nature of interlimb differences has been clearly demonstrated by previous studies by Bishop and colleagues [9,10,49]. Here we expand previous literature demonstrating that, even when the mechanical constraints and contraction modality are the same, i.e., in isometric conditions, analyzing different muscle performance metrics may favour one side or the other. This can be clearly seen in the Figure 2B, which shows the individual pattern of the first eight participants of the sample. In most participants, interlimb asymmetry favours either the dominant or non-dominant limb depending on muscle group and performance metric. Indeed, the asymmetry indices are sparse below and above the zero line, which would indicate perfect symmetry (Figure 2B). These results suggest that most people do not present one limb overall more performative than the other. Conversely, each limb is more performative in one performance metric, independent of dominance.

From a practical point of view, the present findings suggest that to determine strength asymmetries, practitioners should adopt that specific test for each relevant muscle group. Even more importantly, the current study may inform the practitioners when they try to treat strength asymmetries, i.e., when they prescribe physical training to diminish the asymmetry level of a person. The strength asymmetry should not be treated just by performing more exercise on one side (the weakest limb) compared to the other (the strongest limb). Each side may need more specific training for the physical characteristic where it is less proficient. For example, one side may need more maximal strength training while the contralateral side may need more explosive training.

The novel findings of the present study do not come without limitations. First, we measured only two of the many muscle groups of the upper limb (elbow flexors and extensors). Therefore, our results may not translate to shoulder or wrist muscle groups. Second, we only adopted isometric contractions; thus, the asymmetry agreement among other contraction modalities remains unexplored. Third, we only included physically active individuals, therefore, our results do not necessarily transfer to sedentary people or highly trained athletes. Even more importantly, our results do not relate to people participating in highly asymmetric sports such as tennis, badminton, or fencing. Indeed, we expect that the asymmetry levels in asymmetric sports would be much broader than the those reported in the present study. Lastly, we could not compare left- vs. right-handed individuals because we only recruited five left-handed individuals; therefore, this comparison’s statistical power would be too low. However, investigating whether left-hand individuals would show different asymmetry agreements would be attractive.

Future studies should address two main questions that remain open from the present study. First, the neuromuscular determinants of the asymmetry should be elucidated: it is unknown if MVF and RFD asymmetries were more related to central (i.e., neural activation) or peripheral (i.e., muscle size and architecture) characteristics. Second, as we determined the asymmetry only once for each subject, we do not know if those asymmetries fluctuate with time or if they remain stable over long periods.

## 5. Conclusions

At the individual level, the asymmetries are muscle-specific and rarely favour the same limb across different muscle groups. When adopting various performance metrics, there is no one limb more performative than the other in general: each limb is favoured depending on muscle group and performance metric, independent of dominance.

## Figures and Tables

**Figure 1 ijerph-19-08495-f001:**
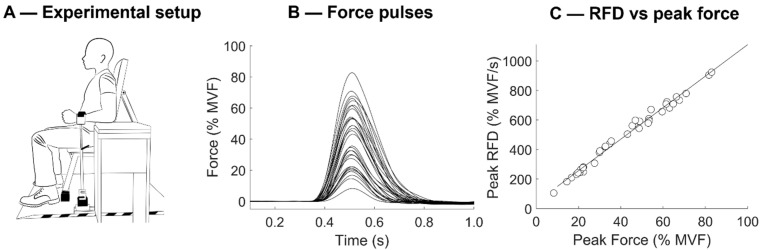
(**A**) The experimental setup adopted to test the elbow flexors and extensors. The arm was maintained in a neutral position. The load cell, which was rigidly attached to the height-adjustable support, allowed mechanical recording in both traction and compression. (**B**) Traces recorded during the execution of the RFD-SF (rate of force development scaling factor) protocol for a representative participant. Left panel: superimposed force traces are reported for each rapid muscle contraction executed at various submaximal amplitudes compared to the maximal voluntary force (MVF). (**C**) Scatterplot representing the peak force and peak RFD of each muscle contraction reported in the right panel. The slope of the linear regression represents the RFD-SF.

**Figure 2 ijerph-19-08495-f002:**
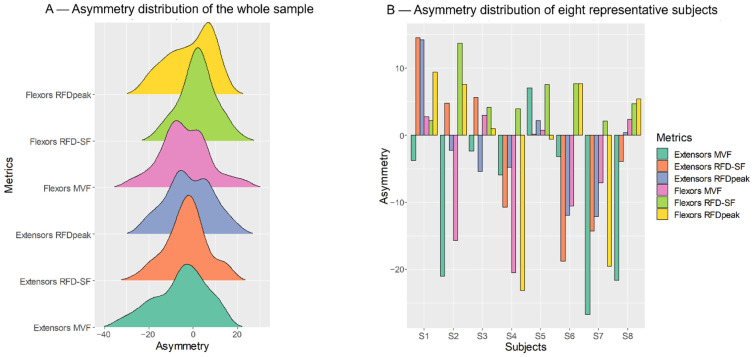
(**A**) Distributions of each performance metric for elbow flexors and extensors. Positive values denote the favour of the dominant limb. As can be seen, the distributions are widely distributed both towards the dominant and non-dominant sides. (**B**) Individual values of bilateral asymmetry indices are reported for each performance metric of the first eight subjects of the sample group. MVF, maximal voluntary force; RFDpeak, peak rate of force development; and RFD-SF (rate of force development scaling factor).

**Table 1 ijerph-19-08495-t001:** The asymmetry index was calculated as ((dominant limb − non-dominant limb)/(dominant limb + non-dominant limb)) × 100 according to previously published studies [34]. Therefore, negative values indicate a favour of non-dominant limb. The percentage of participants favouring non-dominant/symmetric/favouring dominant are computed based on the smallest worthwhile change (SWC).

	Flexors				Extensors			
	Dominant	Non-Dominant	Bilateral Asymmetry Index (%)	Participant Favouring Non-Dominant/Symmetric/Favouring Dominant (%)	Dominant	Non-Dominant	Bilateral Asymmetry Index (%)	Participant Favouring Non-Dominant/Symmetric/Favouring Dominant (%)
MVF (N)	311 ± 118	346 ± 135	−4 ± 11	56/26/19	229 ± 69	258 ± 91	−6 ± 12	41/41/19
RFDpeak (N/s)	4419 ± 1530	4664 ± 1782	−1 ± 10	39/25/37	2970 ± 1019	3050 ± 974	−2 ± 10	9/82/9
RFD-SF (1/s)	9.1 ± 1.4	8.7 ± 1.4	2 ± 8	29/18/54	9.2 ± 1.5	9.8 ± 1.5	−3 ± 9	57/16/27

## Data Availability

Data are available contacting the corresponding author.

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
