# Peer review of "Strength Asymmetries Are Muscle-Specific and Metric-Dependent"

_ijerph, 2022, doi:10.3390/ijerph19148495_

Round 1
Reviewer 1 Report
I have carefully read the manuscript and my opinion is that the manuscript has a merit to be published in your reputable journal with some minor corrections. The manuscript is original, informative and readable. The authors tried to answer three research question: they firstly aimed to examine if dominance affected muscle function, secondly, they investigated if muscle function asymmetries are muscle-specific or, conversely, if one side is overall more performative than the other independently of the muscle group, as well as thirdly, they aimed to investigate if, within each muscle group, asymmetry direction is consistent among various muscle performance metrics (MVF, RFD, and RFD-SF). To answer the first experimental question, they adopted a linear mixed-effects model analysis, while to answer the second and third experimental questions, they tested the agreement between asymmetry direction among heterologous muscle groups and various performance metrics. Overall, the authors concluded the asymmetries are muscle-specific and rarely favor the same side across different muscle performance metrics, while no one side is more performative than the other at the individual level (each limb is favored depending on muscle group and performance metric). The structure of the abstract should be in the following order: the purpose of the study, method, results and short discussion with conclusions. On the other hand, I would appreciate if the authors more carefully describe the subject of the study as I prefer to hear more information about the selected individuals and the methodology of its inclusion into the sample. The authors did not justify the representativeness with “young physically active individuals” from my point of view. I would like to see explanation why the authors selected this subject. At the end, I would also recommend to the authors to prepare the conclusion part in the following order: the main conclusions, the limitations of the study (more precisely) as well as recommendations for the further studies (it is very important to briefly elaborate it and highlight the most important notes. Lastly, I would recommend you to accept this manuscript as soon as it is revised for the further production.
Author Response
We warmly thank this reviewer for their comments and for appreciating the study.
This and others reviewer requested us to more carefully explain the inclusion criteria of the study and the characteristics of the participants. We completely agree with this request and revised the method section accordingly.
We also restructured the limitations section and we included the recommendations for future studies as suggested.
Thanks again for the constructive inputs to our study.
For more point-to-point responses, please see attachment.

Reviewer 2 Report
Thank you for the opportunity to review the manuscript titled “Strength asymmetries are muscle-specific and metric-dependent”. The study provides the measurement of three muscle-performance metrics (MVF, RFDpeak, and RFD-SF) in two muscle groups of the upper limbs (elbow flexors and extensors) of the dominant and non-dominant sides. In Introduction section the authors justify the focus of study in upper limbs cause they play a critical role in everyday living and sports context in non-disabled, amputee, and wheelchair users. It is an interesting and well written article, with figures and tables that help in understanding the results. They concluded that the asymmetries are muscle-specific and rarely favour the same limb across different muscle groups and each limb is favoured depending on muscle group and performance metric, independently of dominance.
Finally, although the discussion is good, it would be interesting to improve this section with the possible exemplification of the application of these results and the “kappa” on the critical role in everyday life and in the sports context in non-disabled people, amputees and wheelchair users, as mentioned in the introduction.
Author Response
We thanks the reviewer for the appreciation of our work. We completely agree with him/her that the application of the study’s findings were not clearly stated in the discussion. For this reason we included a paragraph describing the practical application of our study in the context of everyday life and sports.
For more point-to-point responses, please see attachment.

Reviewer 3 Report
Please see my attached comments and suggestions.
Review
First, I would like to recognize the authors for the data they collected to examine the effect of dominance on the upper limbs' muscle function and the level of agreement in asymmetry direction across various muscle function metrics of two heterologous muscle groups.
The title is clear, concise, and informative.
The abstract is clear and includes the objectives, design, methods, variables considered, main results and most relevant conclusion. The only issue I can identify is that we do not have a clear indication as to who can use these results. Who can use these results and why? I would probably read this study as I am interested in asymmetries in an athletic population, but you may want to add a short statement stating that "these results can be used by…..".
The introduction is clear and follows a logical sequence while all the relevant scientific support is provided. Furthermore, the objectives are clearly set out. In the instruction, you stated that you focused on upper limbs (elbow flexors and extensors) because they play a critical role in everyday living and sports context in non-disabled, amputee, and wheelchair users. The 55 healthy adults are not athletes etc., however. Please note that your results could have been completely different if you included athletes of different sports or wheelchair users; therefore, I do not know if your results can be applied to athletes or wheelchair users. Again, using the upper limbs is a great idea, as well as identifying asymmetries, etc. However, you need to state in your limitations section that these results are derived from the testing of healthy individuals and may not necessarily apply to athletes.
Materials and Methods section
"A total of 55 young (mean age = 30±7 years; 24 females) physically active individuals (body mass = 70±9 kg, body height = 1.74±0.17 m) were recruited for the study". You need to add the information of the male participants as well here. Is the 30±7 years the mean age of the females or the total sample? You
may want to say (24 females and 31 males; mean age = 30±7 years). I know it seems obvious, but it is more apparent to the reader.
Results
The results are presented and correspond to the data obtained. Furthermore, the results provide relevant information in terms of the study's objectives
LINE 248: Table 1 - The asymmetry index was calculated as ((dominant limb − non-dominant limb) / (dominant limb + non-dominant limb)) × 100 according to [34]. Instead of saying according to [34], you may want to say according to Kobayashi and Colleagues or according to previously published studies [34].
Discussion
Overall, I find the results of this study significant, and I would like to see them published. However, you should state that these results may be different when testing athletes of different sports. For example, do you think your results would be the same for tennis players? It would be very interesting to have a future study (replicate your study) on basketball, volleyball and tennis players.
Author Response
We thanks this reviewer for asking us to clearly describe the potential usefulness of our finding. We added a sentence at the end of the abstract with this aim.
We completely agree with this reviewer: our findings are specific to the population that we investigate. For this reason, we expanded the limitation sections by stating that our results should not be applied to other populations, like tennis players and fencers.
We rephrased two sentences in the method and in the results sections as suggested by the reviewer.
For more point-to-point responses, please see attachment.

Reviewer 4 Report
Dear Authors, It was a great pleasure to review your valuable study. Attached my report with this mail. I hope my suggestions and critiques would therefore help to improve the quality of your publications.
Title: Specific topic: The current title “Strength asymmetries are muscle-specific and metric-dependent “ would be more interesting if the authors could modify it by adding the key points such as Isometric/ dominant and no-dominant elbow flexor and extensor muscle group specific /metric dependent tests. Novelty & originality: Good
Abstract: Line-29: Add a conclusive statement on clinical significance of the current study results.
Introduction: Line-88-94: Add some justification on choosing elbow muscle groups for this study.
Methods: Line-107: Sample size/ sampling method missing; add Male: Female ratio as 31:24 Line-108: Add BMI to understand the subject physique Line-111: About their occupation/past level of activity & physical training Line-148: Randomization procedure missing
IRB approval: mentioned Duration & place of the study: missing
Results & Data analysis: Picture/graphs quality need to improve
Discussion: Line-295: Give some neuromusculoskeletal reasoning for dominant and non-dominant difference Line-308: Reasoning for elbow flexors Vs Extensor difference
Conclusion: satisfactory Limitation of study/areas for future research/ nutshell with clinical implication
COI/Fund statement: stated References: Are latest
Author Response
As suggested by this and other reviewers, we added a sentence at the end of the abstract explaining the clinical significance of the study.
We included the subjects' information that were missing. We also explained that the order of the tests was randomized in counterbalanced order.
We also expanded the limitation section and the future study section in the relevant part of the discussion as suggested by this reviewer.
We also expanded the reasoning of the dominant vs non-dominant side differences.
For more point-to-point responses, please see attachment.

Round 2
Reviewer 3 Report
I believe the manuscript has been sufficiently improved to warrant publication in IJERPH.